# Amortized Eigendecomposition for Neural Networks

**Tianbo Li**[1,*], **Zekun Shi**[1,2], **Jiaxi Zhao**[3], **Min Lin**[1]
[1]SEA AI Lab
[2] School of Computing, National University of Singapore
[3] Department of Mathematics, National University of Singapore
*litb@sea.com

## Abstract

Performing eigendecomposition during neural network training is essential for tasks such as dimensionality reduction, network compression, image denoising, and graph learning. However, eigendecomposition is computationally expensive as it is orders of magnitude slower than other neural network operations. To address this challenge, we propose a novel approach called "amortized eigendecomposition" that relaxes the exact eigendecomposition by introducing an additional loss term called eigen loss. Our approach offers significant speed improvements by replacing the computationally expensive eigendecomposition with a more affordable QR decomposition at each iteration. Theoretical analysis guarantees that the desired eigenpair is attained as optima of the eigen loss. Empirical studies on nuclear norm regularization, latent-space principal component analysis, and graphs adversarial learning demonstrate significant improvements in training efficiency while producing nearly identical outcomes to conventional approaches. This novel methodology promises to integrate eigendecomposition efficiently into neural network training, overcoming existing computational challenges and unlocking new potential for advanced deep learning applications.

## 1 Introduction

Eigendecomposition is a fundamental technique in linear algebra that finds applications across numerous scientific domains ranging from quantum many-body problems to multivariate statistical analysis. In the context of deep learning, eigendecomposition also plays a crucial role in tasks such as weights normalization [8, 16, 41], dimensionality reduction [6, 44, 27, 38], network compression [17, 30], image denoising [13, 12, 14], graph adversarial learning [10, 18, 47]. By uncovering the structure of networks, eigendecomposition allows us to enforce low-rankness, ensuring generalization, robustness, and computational efficiency. Eigendecomposition is also instrumental in the spectral analysis of graphs, where it can detect community structure, which is essential in spectral graph neural networks. The ability of eigendecomposition to detect the intrinsic matrix structures and properties makes it a valuable tool in various machine learning tasks with neural networks.

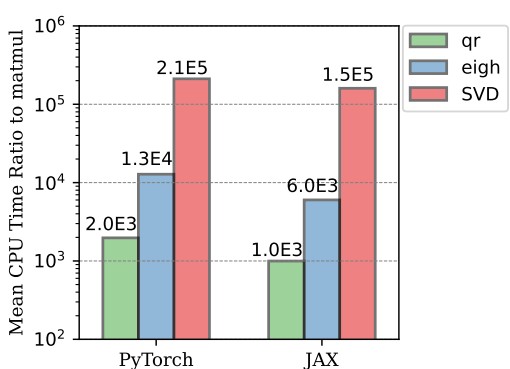

Figure 1: A comparison that illustrates the forward execution time of three linear algebra operations: `qr`, `eigh`, and `svd`, when performed on a $10000 \times 10000$ matrix using PyTorch and JAX. The presented values represent the mean ratios of the execution time relative to that of matrix multiplication (`matmul`) of 100 runs.

38th Conference on Neural Information Processing Systems (NeurIPS 2024).

Despite its straightforward definition, the computation of eigenvalues is quite challenging. Eigendecomposition algorithms are inherently iterative, involving a sequence of expensive operations such as Arnoldi iteration, QR iteration, and Rayleigh quotient Iteration [39]. Additionally, it usually takes thousands of iterations to reach a desirable tolerance level. For instance, the locally optimal block preconditioned conjugate gradient (LOBPCG) method [22]–a widely-used eigenvalue solver–often requires dozens to hundreds of iterations to achieve convergence [22, 9]. Figure 1 presents a comparative analysis of the execution times for eigendecomposition and other computational operations using PyTorch [33] and JAX [2]. The figure shows that the execution speeds for `eigh` and `svd` are remarkably slower—by 4 to 5 orders of magnitude—relative to matrix multiplication. This substantial disparity in execution speed indicates operations such as `eigh` and `svd`, once used, will be the bottleneck of computation cost. Conversely, the QR decomposition, often employed for orthogonality, is considerably less computationally expensive. This observation has motivated us to explore the possibility of reducing the iterative computation of eigendecomposition with lower-cost operations.

When eigendecomposition is incorporated into the training of a neural network, a nested loop scenario arises, where eigendecomposition acts as the inner loop and the neural network's loss minimization serves as the outer loop. Notably, it is not always easy to be aware of this inner loop of eigendecomposition, as it is encapsulated by the high-level functions provided within deep learning frameworks. However, in this context, this inner loop does not require full convergence during each iteration, given that the remaining parameters have not reached optima. The inner loop can be relaxed and optimized jointly with the training loss, allowing for a more flexible and efficient training process. The key idea can be summarized as follows:

*Eigendecomposition within a neural network does not have to reach full convergence during each training step; it simply needs to contribute to the desired outcome by the end of the training process.*

In this paper, we present a novel approach named "amortized eigendecomposition" for training neural networks that require eigenvalues or eigenvectors. Instead of using computationally expensive eigendecomposition decoupled with the training of neural networks, we proposed to relax it into an unconstrained optimization problem on the Stiefel manifold by adding an eigen loss. This relaxation only requires a QR decomposition at each iteration, thus is more efficient. Moreover, through empirical observations, we have found that although the relaxed optimization problem with eigen loss does not involve eigendecomposition in every iteration, the amortized optimization approach consistently achieves the desired results. It achieves nearly identical performance to traditional methods but with significantly improved speed.

## 2 Eigendecomposition in Neural Networks

In this paper, we consider a general class of neural networks that incorporate eigendecomposition. We formulate this family of problems as a constrained optimization problem:

$$\min_{\boldsymbol{\theta}} f\big(h_{\boldsymbol{\theta}}(\boldsymbol{X}), \boldsymbol{V}, \boldsymbol{\Lambda}\big), \qquad \text{s.t. } \boldsymbol{V}^{\top}\boldsymbol{\Lambda}\boldsymbol{V} = \boldsymbol{A} \tag{1}$$

Here, the encoder $h_{\boldsymbol{\theta}}$ maps the data $\boldsymbol{X} \in \mathbb{R}^{n \times p}$ into a latent space. In addition to the latent representation, the loss function $f$ also incorporates the eigenpair $\boldsymbol{\Lambda}$ and $\boldsymbol{V}$ of a symmetric matrix $\boldsymbol{A}$. The matrix $\boldsymbol{A}$ can be constructed from $h_{\boldsymbol{\theta}}(\boldsymbol{X})$, such as a covariance matrix or a similarity matrix, or it can depend solely on the parameters. Notably, $\boldsymbol{A}$ is subject to changes during network training due to its dependency on $\boldsymbol{\theta}$. The computational graph for each iteration of such model structure can be written as,

$$\underbrace{\boldsymbol{X} \longrightarrow h_{\boldsymbol{\theta}}(\boldsymbol{X})}_{\text{encoding}} \longrightarrow \underbrace{\boldsymbol{A} \longrightarrow (\boldsymbol{V}, \boldsymbol{\Lambda})}_{\text{eigendecomposition}} \longrightarrow \underbrace{f\big(h_{\boldsymbol{\theta}}(\boldsymbol{X}), \boldsymbol{V}, \boldsymbol{\Lambda}\big)}_{\text{downstream task}}. \tag{2}$$

The corresponding algorithm of the above computational graph is shown in Algorithm 1. Solving such problems, however, is computationally expensive, since it requires preserving the eigendecomposition constraint of $\boldsymbol{A}$ after each update of $\boldsymbol{\theta}$. This structure encompasses a wide range of learning problems. We present several representative examples and investigate them in our numerical experiments next.

**Nuclear Norm Regularization** The nuclear norm of a matrix is defined as the sum its the singular values. Due to its convexity, regularization via the nuclear norm is employed to encourage low-rank

structures within the learned parameters. This approach proves beneficial in a variety of applications, such as matrix completion [5, 4], image denoising [13, 12, 14, 45]. Furthermore, eigendecomposition and singular value decomposition are also used for pruning or compressing neural networks, by decomposing the weight matrices of the network and approximating the original network with fewer parameters [17, 30]. The objective function of this type of problem can be written as,

$$\min_{\boldsymbol{\theta}} f\big(h_{\boldsymbol{\theta}}(\boldsymbol{X})\big) + \eta\|\boldsymbol{\theta}\|_* \tag{3}$$

Here, $\eta$ is a regularization coefficient that controls the rank of the parameter matrix. In classical methods, the nuclear norm $\|\boldsymbol{\theta}\|_*$ is usually calculated via singular value decomposition.

**Whitening in Neural Networks**  Whitening is a transformation technique extensively utilized in neural networks to standardize features by ensuring they have zero mean and unit variance, and are decorrelated from each other. In neural network applications, whitening can be categorized into parameter-space whitening and feature-space whitening. Parameter-space whitening, often achieved through PCA/ZCA, is a prevalent method applied during neural network training to improve stability and accelerate convergence [37, 41, 16, 8]. Feature-space whitening, in contrast, applies PCA to the intermediate representations within the network. This process aligns features with the axes of greatest variance, thereby facilitating dimensionality reduction [15, 42, 34, 25]. A concrete example of feature-space whitening is its incorporation into an auto-encoder architecture, which can be expressed mathematically as:

$$\min_{\boldsymbol{\omega},\boldsymbol{\theta}} \left\|\mathsf{Dec}_{\boldsymbol{\omega}}\Big(\boldsymbol{V}\boldsymbol{V}^{\mathsf{T}}\mathsf{Enc}_{\boldsymbol{\theta}}(\boldsymbol{X})\Big) - \boldsymbol{X}\right\|_{\mathrm{F}} \tag{4}$$

where $\boldsymbol{V}$ is spanned by the first several largest eigenvectors of the covariance of $\mathsf{Enc}_{\boldsymbol{\theta}}(\boldsymbol{X})$ and $\boldsymbol{\theta}, \boldsymbol{\omega}$ represent the parameters of the decoder and encoder, respectively.

**Graph Structure Learning**  Graph structure learning is an area of machine learning that aims to deduce the latent structure of a graph or network from observed data [23]. This domain has significant applications in graph adversarial learning, which seeks to bolster the robustness of graph neural networks against adversarial attacks [10, 18, 47]. In such contexts, the adjacency matrix is usually often compromised by adversarial modifications while feature matrix $\boldsymbol{X}$ remains unaffected. The primary objective is to learn both a clean graph structure and perform accurate node classification. From this perspective, we follow the approach proposed in [18] and the objective function can be formulated as:

$$\min_{\boldsymbol{\theta},\widehat{\boldsymbol{L}}} \left\|\mathsf{GNN}_{\boldsymbol{\theta}}(\boldsymbol{X},\widehat{\boldsymbol{L}}) - \boldsymbol{Y}\right\|_2 + \alpha\big\|\widehat{\boldsymbol{L}}\big\|_* + \beta\big\|\widehat{\boldsymbol{L}} - \boldsymbol{L}\big\|_{\mathrm{F}}^2 \tag{5}$$

Here $\widehat{\boldsymbol{L}}$ represents a low-rank Laplacian matrix that approximates the original graph Laplacian $\boldsymbol{L}$. The three parts of the loss function correspond to the node classification loss, low-rank constraint, and unnoticeable adversarial attacks, respectively. However, this approach relies on singular value decomposition at every iteration, which is computationally prohibitive for large-scale networks.

## 3  Differentiable Optimization for Eigendecomposition

Before introducing the proposed method, let's begin by examining a more straightforward case: determining the eigenvalues of a symmetric matrix $\boldsymbol{A} \in \mathbb{R}^{n \times n}$ through constrained optimization.[1] The eigenvalues of $\boldsymbol{A}$ are denoted as $\lambda_1 \geqslant \lambda_2 \geqslant \cdots \geqslant \lambda_n$. The largest eigenvalue of $\boldsymbol{A}$ can be obtained by directly maximizing the *Rayleigh quotient*:

$$\lambda_1 = \max_{\boldsymbol{u} \in \mathbb{R}^n} \frac{\boldsymbol{u}^{\mathsf{T}}\boldsymbol{A}\boldsymbol{u}}{\boldsymbol{u}^{\mathsf{T}}\boldsymbol{u}}. \tag{6}$$

where the maximum value corresponds to the largest eigenvalue and the associated eigenvector is given by the normalized $\boldsymbol{u}$ at the optimum. This method can be generalized to identifying *the optimal subspace*, which is a common problem in dimensionality reduction and feature selection contexts. The optimal subspace for a symmetric matrix is defined as the subspace spanned by its $k$ largest eigenvectors. This can be found by solving the constrained optimization problem:

$$\max_{\boldsymbol{U} \in \mathbb{U}^{n \times k}} \quad \mathsf{tr}(\boldsymbol{U}^{\mathsf{T}}\boldsymbol{A}\boldsymbol{U}) \tag{7}$$

---

[1]Although the findings in this paper can be readily extended to the complex space, we specifically focus on the real space throughout our study.

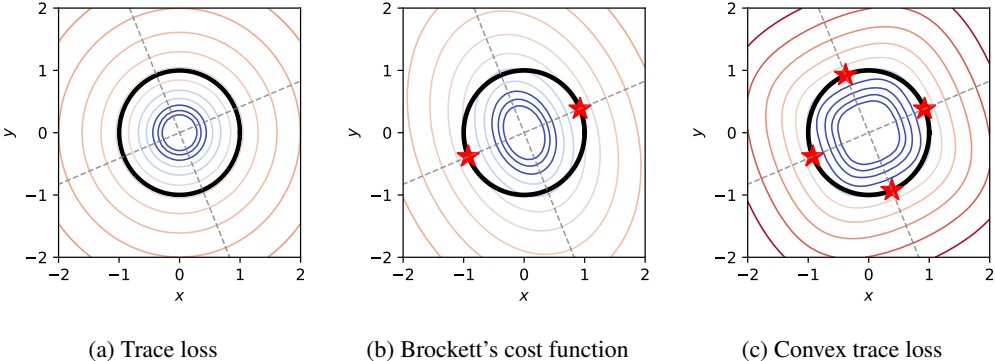

|  (a) Trace loss | (b) Brockett's cost function | (c) Convex trace loss |

Figure 2: An illustration on disrupting the rotational symmetry of the trace loss. We aim to solve eigenvectors for a 2-dimensional symmetric matrix $\boldsymbol{A} = \begin{pmatrix} 0.8 & 0.2 \\ 0.2 & 0.4 \end{pmatrix}$ with three loss functions. We parameterize an orthonormal matrix $\boldsymbol{U} = \begin{pmatrix} x & y \\ -y & x \end{pmatrix}$, which is subject to the constraint $x^2 + y^2 = 1$. The plot displays the contours of the landscapes of three different loss functions as they vary with $x$ and $y$: (a) trace loss $\mathsf{tr}(\boldsymbol{U}\boldsymbol{A}\boldsymbol{U}^\top)$, (b) Brockett's cost function $\mathsf{tr}(\boldsymbol{M}\boldsymbol{U}\boldsymbol{A}\boldsymbol{U}^\top)$ where $\boldsymbol{M} = \mathsf{diag}(1, 0.2)$ and (c) convex loss function $\mathsf{tr}(f(\boldsymbol{U}\boldsymbol{A}\boldsymbol{U}^\top))$ where $f$ is an element-wise exponential function. The feasible area of the constraint is depicted with a black circle. The red stars signify the optima of the loss in the feasible area. The dashed grey lines represent the true eigenvector direction of $\boldsymbol{A}$. We see that, the trace loss results in infinitely many optimal solutions due to its rotational symmetry. In contrast, both Brockett's cost function and the convex loss function reshape the optimization landscape, breaking this symmetry and leading to the identification of the correct eigenvectors.

where the Stiefel manifold $\mathbb{U}^{n\times k} = \{\boldsymbol{U} \in \mathbb{R}^{n\times k} \mid \boldsymbol{U}^\top\boldsymbol{U} = \boldsymbol{I}\}$ is the set of $n \times k$ matrices with orthonormal columns. Maximizing this trace function yields the optimal subspace spanned by the column vectors of $\boldsymbol{U}$.

**Rotational Symmetry**   It's important to recognize, however, that the optimal $\boldsymbol{U}$ of the above optimization problem does NOT represent the eigenvectors of $\boldsymbol{A}$. This is due to the rotational symmetry of the trace function that for any orthonormal matrix $\boldsymbol{C} \in \mathbb{U}^{k\times k}$, the equation $\mathsf{tr}(\boldsymbol{U}^\top\boldsymbol{A}\boldsymbol{U}) = \mathsf{tr}(\boldsymbol{C}^\top\boldsymbol{U}^\top\boldsymbol{A}\boldsymbol{U}\boldsymbol{C})$ always holds. As a result, there are an infinite number of solutions to the optimization problem Eq. (7), all spanning the same subspace as the desired sets of eigenvectors. A visual illustration is provided in Figure 2a.

To accurately obtain the eigenvalues and eigenvectors, it is necessary to refine the traditional trace loss function. This paper introduces two approaches for achieving the correct eigendecomposition. The first method utilizes Brockett's cost function, which applies distinct weights to the diagonal elements of the matrix product $\boldsymbol{U}^\top\boldsymbol{A}\boldsymbol{U}$, effectively differentiating the importance of each eigenvalue. The second method involves applying an element-wise convex function directly to the diagonal elements, resulting in an exact eigendecomposition. We will now elaborate on these methods.

**The Brockett's Cost Function**   The first modification made to the trace loss is Brockett's cost function [3]. It enables the extraction of eigenvectors and eigenvalues by solving the following optimization problem:

$$\max_{\boldsymbol{U}\in\mathbb{U}^{n\times k}} \quad \mathsf{tr}(\boldsymbol{M}\boldsymbol{U}^\top\boldsymbol{A}\boldsymbol{U}). \tag{8}$$

Here $\boldsymbol{M} = \mathsf{diag}(m_1, m_2, \cdots, m_k) \in \mathbb{R}^{k\times k}$ is a diagonal weight matrix with distinct diagonal elements. The matrix $\boldsymbol{M}$ is structured such that all its elements are distinct numbers. Specifically, we can denote these elements as $0 < m_1 < m_2 < \cdots < m_k$. This ordering allows $\boldsymbol{M}$ to assign distinct weights to the diagonal elements of the product $\boldsymbol{U}^\top\boldsymbol{A}\boldsymbol{U}$, effectively disrupting the rotational invariance inherent in the trace loss and thus enabling the determination of eigenvalues and eigenvectors through optimization. In fact, $\boldsymbol{M}$ can be any diagonal matrix with distinct diagonal

elements. A practical choice could be $m_i = i/k$ or $m_i = i$ as suggested by [3]. The next theorem illustrates why this approach can yield the exact eigendecomposition.

**Theorem 1** (Trace inequality with weight matrix). *Consider a $n$-dimensional symmetric matrix $\boldsymbol{A}$. Let $\boldsymbol{M}$ be a $k$-dimensional diagonal matrix with elements $0 < m_1 < m_2 \cdots < m_k$. For an arbitrary matrix $\boldsymbol{U} \in \mathbb{U}^{n \times k}$ with orthogonal columns, the following inequality always holds:*

$$\sum_{i=1}^{k} m_{k-i+1}\lambda_i \leqslant \mathsf{tr}(\boldsymbol{M}\boldsymbol{U}^{\mathsf{T}}\boldsymbol{A}\boldsymbol{U}) \leqslant \sum_{i=1}^{k} m_i\lambda_i. \tag{9}$$

*The equalities are achieved if and only if $\boldsymbol{U}$ contains the eigenvectors corresponding to the $k$ largest (smallest) eigenvalues of $\boldsymbol{A}$.*

This approach can be viewed as an extension of the trace inequalities originally put forward by Von Neumann [40, 31] and further developed by Ruhe [35]. A detailed demonstration is available in the seminal work by Brockett [3]. Additionally, we offer an alternative proof leveraging the Cauchy–Schwarz inequality, which is presented in the Appendix for reference. A more generalized version of this trace inequality is discussed in Liang et al. [26].

**The Convex Trace Loss**   The second method employs a strictly monotonic convex function $f$, applied element-wise to the diagonal components of $\boldsymbol{U}^{\mathsf{T}}\boldsymbol{A}\boldsymbol{U}$. This perturbation also disrupts the rotational symmetry inherent in the trace loss. The convex nature of $f$ alters the curvature of the loss landscape, thereby ensuring a unique optimal solution corresponding to the eigenvectors. The convex trace loss function, aimed at extracting the $k$ largest eigenvalues, is expressed as:

$$\max_{\boldsymbol{U} \in \mathbb{U}^{n \times k}} \quad \mathsf{tr}f(\boldsymbol{U}^{\mathsf{T}}\boldsymbol{A}\boldsymbol{U}) \tag{10}$$

The optimal $\boldsymbol{U}_*$ that achieves the maximum in the above objective are the eigenvectors corresponding to the $k$ largest eigenvalues, and the eigenvalues can be obtained by the diagonal elements of $\boldsymbol{U}_*^{\mathsf{T}}\boldsymbol{A}\boldsymbol{U}_*$. The next theorem provides a formal validation of this assertion.

**Theorem 2** (Trace inequality with convex function). *Let $\boldsymbol{A}$ be a given n-dimensional symmetric matrix and let $\boldsymbol{U}$ be a matrix of size $n \times k$ that resides on the Stiefel manifold. Suppose $f : \mathbb{R} \to \mathbb{R}$ be a monotonically increasing, convex function applied element-wise. The following inequalities hold:*

$$k\left(f\left(\frac{1}{k}\mathsf{tr}(\boldsymbol{A})\right)\right) \leqslant \mathsf{tr}\left(f\left(\boldsymbol{U}^{\mathsf{T}}\boldsymbol{A}\boldsymbol{U}\right)\right) \leqslant \sum_{i=1}^{k} f(\lambda_i), \tag{11}$$

*The rightmost inequality becomes an equality if and only if $\boldsymbol{U}$ comprises the eigenvectors of $\boldsymbol{A}$ that correspond to the $k$ largest eigenvalues, The leftmost inequality is met with equality when $\boldsymbol{U}$ is such that all diagonal elements of the matrix $\boldsymbol{U}^{\mathsf{T}}\boldsymbol{A}\boldsymbol{U}$ are equal.*

This theorem is established through the application of Jensen's inequality. The detailed proof is provided in the Appendix B.3 for reference. In the above objective, suitable choices of $f$ include: $f(x) = \exp(x)$ on $\mathbb{R}$; $f(x) = x^\alpha$ on $\mathbb{R}^+$ where $\alpha > 1$; and $f(x) = \tan(x)$ on $[0, \pi/2]$. For finding the $k$ smallest eigenvalues, a simple modification can be made by replacing the function $f$ with an element-wise monotonically increasing concave function and then minimizing the trace loss. A visual representation of the optimization process of the three trace losses is presented in Figure 2.

## 4   The Amortized Eigendecomposition Approach

The proposed amortized eigendecomposition approach aims to modify the eigendecomposition operation within the neural network's computational graph, as illustrated in Eq. (2). This replacement involves two steps: First, the set of eigenvectors form a matrix on the Stiefel manifold reparameterized through computationally efficient operations such as QR decomposition. Then, the loss function is adjusted to ensure that its optimal solutions precisely correspond to the eigendecomposition. The computational graph for this amortized eigendecomposition method is formulated as follows:

$$\underbrace{\boldsymbol{X} \longrightarrow h_{\boldsymbol{\theta}}(\boldsymbol{X})}_{\text{encoder}} \longrightarrow \boldsymbol{A} \longrightarrow \underbrace{f_{\boldsymbol{\omega}}\big(h_{\boldsymbol{\theta}}(\boldsymbol{X}), \boldsymbol{U}, \boldsymbol{\Sigma}\big)}_{\text{model loss}} + \eta \underbrace{g\,(\boldsymbol{U}, \boldsymbol{A})}_{\text{eigen loss}} \longleftarrow \underbrace{\boldsymbol{U} \xleftarrow{\text{QR}} \boldsymbol{W}}_{\text{reparameterization}}. \tag{12}$$

In the computation graph, the eigendecomposition operation is circumvented and substituted with a more efficient QR operation. The QR operation is employed to reparameterize the orthogonal matrix

| **Algorithm 1** The conventional eigendecomposition in a neural network outlined in Eq. (2) | **Algorithm 2** The amortized eigendecomposition technique outlined in Eq. (12) |
|---|---|
| **Input:** Dataset $\boldsymbol{X}$, encoder $h_{\boldsymbol{\theta}}$, task $f_{\boldsymbol{\omega}}$; | **Input:** Dataset $\boldsymbol{X}$, encoder $h_{\boldsymbol{\theta}}$, task $f_{\boldsymbol{\omega}}$, and $\eta$; |
| 1: Initialize model parameter $\boldsymbol{\theta}$ and $\boldsymbol{\omega}$; | 1: Initialize model parameter $\boldsymbol{\theta}$, $\boldsymbol{\omega}$ and $\boldsymbol{W}$; |
| 2: **while** not converged **do** | 2: **while** not converged **do** |
| 3:   compute $\boldsymbol{A}$ from $h_{\boldsymbol{\theta}}(\boldsymbol{X})$; | 3:   compute $\boldsymbol{A}$ from $h_{\boldsymbol{\theta}}(\boldsymbol{X})$; |
| 4:   $\boldsymbol{V}, \boldsymbol{\Lambda} = \mathsf{eigh}(\boldsymbol{A})$; | 4:   $\boldsymbol{U} = \mathsf{QR}(\boldsymbol{W})$, $\boldsymbol{\Sigma} = \mathsf{diag}(\boldsymbol{U}^\mathsf{T}\boldsymbol{A}\boldsymbol{U})$; |
| 5:   compute $f_{\boldsymbol{\omega}}(h_{\boldsymbol{\theta}}(\boldsymbol{X}), \boldsymbol{V}, \boldsymbol{\Lambda})$; | 5:   compute $f_{\boldsymbol{\omega}}(h_{\boldsymbol{\theta}}(\boldsymbol{X}), \boldsymbol{U}, \boldsymbol{\Sigma}) + \eta g(\boldsymbol{A}, \boldsymbol{U})$; |
| 6:   update $\boldsymbol{\theta}, \boldsymbol{\omega}$ by gradient descent; | 6:   update $\boldsymbol{\theta}, \boldsymbol{\omega}, \boldsymbol{W}$ by gradient descent; |
| 7: **end while** | 7: **end while** |

on the Stiefel Manifold, leading to a substantial acceleration at each iteration. Moreover, instead of forcing the eigendecomposition constraint at each iteration, we relax it to an eigen loss which is jointly optimized with the training loss of the neural network as a nested optimization loop. This training process is outlined in Algorithm 2, where the key difference of our amortized optimization for the eigendecomposition approach is highlighted in red background color.

**Reparameterize the Stiefel Manifold**   There are three prevalent methods for reparameterizing an orthogonal matrix: through the matrix exponential, the Cayley transform, and QR decomposition. Due to the QR decomposition's better numerical stability and efficiency for non-square matrices $\boldsymbol{U}$, we employ it for reparameterizing a matrix with orthonormal columns:

$$\boldsymbol{U} = \mathsf{QR}(\boldsymbol{W}). \tag{13}$$

In this formulation, $\boldsymbol{W}$ is dimensionally consistent with $\boldsymbol{U}$. QR decomposition is more computationally efficient than eigendecomposition and singular value decomposition as shown in Fig. 1. Additionally, the backward computation of the QR decomposition is well-defined and has been efficiently optimized in modern deep learning frameworks, such as PyTorch and JAX. For details on the back-propagation process of the QR decomposition, see [36].

**Relaxation with Eigen Loss**   Previously, we observed that optimizing the Brockett-type or convex trace loss directly enables us to obtain precise eigenvalues and eigenvectors. For any loss function that depends on the eigenvectors or eigenvalues, as specified in Eq.(1), we can transform this loss into a regularized version incorporating the eigen loss. This relaxation allows us to forego the need for explicit eigendecomposition at every iteration, while ultimately achieving equivalent outcomes. We now examine several general scenarios that encompass the majority of cases and explore how to implement this relaxation technique.

In the general case, the model loss in Eq. (1) depends on both the eigenvectors and eigenvalues. This constrained optimization problem can be relaxed by introducing an eigen loss as a regularizer, which is formulated as:

$$\min_{\boldsymbol{\theta}} f\big(h_{\boldsymbol{\theta}}(\boldsymbol{X}), \boldsymbol{V}, \boldsymbol{\Lambda}\big) \xrightarrow[\substack{\textit{largest} \text{ eigenpair}}]{(\boldsymbol{V}, \boldsymbol{\Lambda}) \text{ is the } k} \min_{\boldsymbol{\theta}, \boldsymbol{W}} f\big(h_{\boldsymbol{\theta}}(\boldsymbol{X}), \boldsymbol{U}, \boldsymbol{\Sigma}\big) - \eta\,\mathsf{tr}(\boldsymbol{M}\boldsymbol{U}^\mathsf{T}\boldsymbol{A}_{\boldsymbol{\theta}}\boldsymbol{U}). \tag{14}$$

In this reformulation, $\boldsymbol{U} \in \mathbb{U}^{n \times k}$, is reparameterized via a QR operation as shown in Eq. (13). This relaxation circumvents the need for eigendecomposition at each iteration by using the computationally cheaper QR decomposition, while still ensuring that the optimal solution corresponds to the precise eigendecomposition of the matrix $\boldsymbol{A}_{\boldsymbol{\theta}}$.

The second type of optimization problem involves scenarios where the model loss is independent of the eigenvalues and depends solely on the eigenvectors, such as in the latent-space PCA network expressed in Eq. (4). To enhance the efficiency of the solution process, the problem can be reformulated to include a trace penalty term, such as Brockett's cost, which is given by

$$\min_{\boldsymbol{\theta}} f\big(h_{\boldsymbol{\theta}}(\boldsymbol{X}), \boldsymbol{V}\big) \xrightarrow[\text{eigenvectors}]{\boldsymbol{V} \text{ is } k \textit{ largest}} \min_{\boldsymbol{\theta}, \boldsymbol{W}} f\big(h_{\boldsymbol{\theta}}(\boldsymbol{X}), \boldsymbol{U}\big) - \eta\,\mathsf{tr}(\boldsymbol{M}\boldsymbol{U}^\mathsf{T}\mathsf{StopGrad}(\boldsymbol{A}_{\boldsymbol{\theta}})\boldsymbol{U}). \tag{15}$$

In this formulation, a stop gradient operation is applied to $\boldsymbol{A}$, since the eigen loss involves $\boldsymbol{A}$ which relies on the parameter $\boldsymbol{\theta}$. By introducing the stop gradient operation, we prevent this regularization term from propagating gradients back to $\boldsymbol{\theta}$.

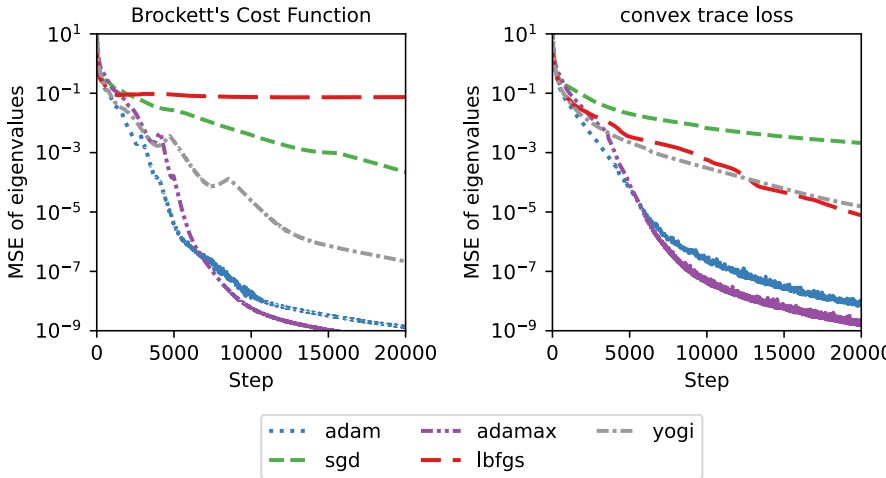

Figure 3: Convergence analysis on finding 50 largest eigenvalues on random $1000 \times 1000$-dimensional symmetric matrices. (a): Convergence curves using Brockett's cost and convex trace loss ($f(x) = x^{1.5}$). (b) The fine-tuning convergence on a series of similar matrices.

Besides the above relaxations, further simplification is possible if the loss function conforms to the structure of Brockett's cost function or a convex trace function. That is, when the model loss $f_\omega$ is a non-uniform linear combination of the eigenvalues, or it is monotonic and convex (or concave) to the eigenvalues, the trace penalty term is unnecessary. The loss function will inherently converge to the correct eigendecomposition. This principle is exemplified in the context of the nuclear norm regularization problem, which will be illustrated later. This can be formulated as:

$$\min_{\boldsymbol{\theta}} f\big(h_{\boldsymbol{\theta}}(\boldsymbol{X}), \boldsymbol{V}, \boldsymbol{\Lambda}\big) \xrightarrow[\text{or a non-uniform linear combination of } \boldsymbol{\Lambda}]{f \text{ is monotonic and concave w.r.t } \boldsymbol{\Lambda}} \min_{\boldsymbol{\theta}, \boldsymbol{W}} f\big(h_{\boldsymbol{\theta}}(\boldsymbol{X}), \boldsymbol{U}, \boldsymbol{\Sigma}\big). \tag{16}$$

## 5 Experiments

In this section, we present an evaluation of our approach, focusing on four specific tasks. Firstly, we demonstrate the convergence properties of our method empirically. Next, we measure the efficacy and efficiency of our amortized eigendecomposition technique over nuclear norm regularized auto-encoder and latent-space PCA using the MNIST dataset. Lastly, we assess the effectiveness of our approach in the context of graph adversarial learning tasks. We implement our approach with the deep learning framework JAX [2]. All the experiments of our approach are conducted on a single NVIDIA A100 GPU with 40GB memory. The two fundamental questions we investigate are as follows:

- *Does our approach accurately identify the eigendecomposition and singular value decomposition?*

- *How does the efficiency of our method compare to that of traditional techniques?*

### 5.1 Convergence

In this experiment, we evaluate the numerical error and convergence speed of our algorithm applied to solving eigendecomposition. We randomly generate ten symmetric matrices of size $1000 \times 1000$. The first 50 eigenvalues of these matrices range from 1 to 50, while the remaining eigenvalues lie between 0 and 1. Our objective is to compute the 50 largest eigenvalues by minimizing Brockett's cost function and convex trace loss (we adopt $f(x) = x^{1.5}$ as the convex function). To achieve this, we employ several optimization algorithms, including Adam [20], Adamax [20], Yogi [46], SGD, and L-BFGS [28]. These algorithms are provided by the Optax and JAX-opt libraries [7]. We measure the mean square error (MSE) of the eigenvalues to the number of training iterations. The results are illustrated in Figure 3.

Table 1: Evaluation of execution times per iteration on three tasks.

| Task | Dimension | Backbone time (s/iter) $t_0$ | Backbone+ eigh/svd time (s/iter) $t_1$ | Backbone+ our method time (s/iter) $t_2$ | Speed-up $\frac{t_1-t_0}{t_2-t_0}$ |
|---|---|---|---|---|---|
| Nuclear norm regularization | $128 \times 128$ | 5.275E-2 | 8.323E-2 | 6.025E-2 | $4.06\times$ |
| | $256 \times 256$ | 5.600E-2 | 1.209E-1 | 6.080E-2 | $13.5\times$ |
| | $512 \times 512$ | 7.186E-2 | 2.616E-1 | 7.366E-2 | $105.4\times$ |
| Latent-space PCA | $256 \times 2$ | 4.178E-3 | 1.446E-2 | 1.117E-2 | $1.47\times$ |
| | $512 \times 2$ | 6.792E-3 | 2.918E-2 | 2.224E-2 | $1.45\times$ |
| | $1028 \times 2$ | 1.434E-2 | 7.018E-2 | 5.467E-2 | $1.39\times$ |
| Low-rank GCN | $2708 \times 16$ | 1.021E-3 | 1.769E-2 | 1.732E-3 | $23.4\times$ |
| | $3312 \times 16$ | 1.367E-3 | 2.825E-2 | 2.498E-3 | $23.7\times$ |
| | $19717 \times 16$ | 1.931E-2 | 4.941E+0 | 2.731E-2 | $615.2\times$ |

This result demonstrates that both loss functions are capable of identifying the correct eigenvalues with a small numerical error of $10^{-9}$. However, there is a noticeable difference in convergence speed. For both trace loss functions, the Adam and Adamax optimizers outperform the others, achieving faster convergence rates. Brockett's cost function, which introduces a linear combination of the trace elements, is more numerically stable compared to the convex trace loss, resulting in faster convergence. This experiment validates the efficiency of the differentiable optimization framework for computing the $k$ largest eigenvalues.

## 5.2 Nuclear Norm Regularization

In this experiment, we apply the amortized eigendecomposition approach to the nuclear norm regularization problem, as outlined in Eq. (3). The experimental framework entails training an auto-encoder on the MNIST dataset by minimizing the reconstruction loss with a nuclear norm regularizer applied to the weight matrix $\boldsymbol{\theta} \in \mathbb{R}^{n \times k}$ of the encoder's last layer. We employ a relaxation technique to the original problem defined in Eq. (16), which can be expressed as:

$$\min_{\boldsymbol{\omega},\boldsymbol{\theta}} \left\| \mathsf{Dec}_{\boldsymbol{\omega}}\big(\mathsf{Enc}_{\boldsymbol{\theta}}(\boldsymbol{X})\big) - \boldsymbol{X} \right\|_{\mathrm{F}} + \eta \|\boldsymbol{\theta}\|_* \longrightarrow \min_{\boldsymbol{\omega},\boldsymbol{\theta},\boldsymbol{W}} \left\| \mathsf{Dec}_{\boldsymbol{\omega}}\big(\mathsf{Enc}_{\boldsymbol{\theta}}(\boldsymbol{X})\big) - \boldsymbol{X} \right\|_{\mathrm{F}} + \eta \sum_{i=1}^{k} \left\| \boldsymbol{\theta}\boldsymbol{u}_i \right\|_2 \quad (17)$$

where $\boldsymbol{u}_i$'s are the orthogonal column vectors of $\boldsymbol{U}$, which are parameterized by $\boldsymbol{W}$ through Eq. (13). The architectures of the encoder and the decoder are constructed as a 2-layer MLP with hidden layer dimensions of D = 128, 256, and 512. For comparison, we also implement the approach based on singular value decomposition (using the svd function). It should be noted that in the current versions of JAX, both the eigh and svd functions are limited to operations on full matrices.

The average execution time per iteration for the baseline backbone with only reconstruction loss, the backbone with svd, i.e. LHS of Eq. (17), and the backbone utilizing amortized eigendecomposition, i.e. RHS of Eq. (17) are reported in Table 1. We denote these execution times as $t_0$, $t_1$, and $t_2$ respectively, and define the speed-up ratio for our approach relative to the svd as:

$$\text{speed-up} = \frac{t_1 - t_0}{t_2 - t_0}. \quad (18)$$

This ratio represents the improvement in execution speed of our eigendecomposition method compared to the standard svd, relative to the baseline backbone performance.

## 5.3 Latent-space Principle Component Analysis

We investigate the effectiveness of our approach for the latent-space PCA method, as described in Eq. (4), using the MNIST dataset. Computing the eigenvectors of the large-scale covariance matrix in each iteration significantly increases the computation overhead, while our method amortizes this cost by jointly minimizing an additional loss. Moreover, we also aim to ensure that the first two eigenvalues are significantly larger than the subsequent ones. In the following objective function, the

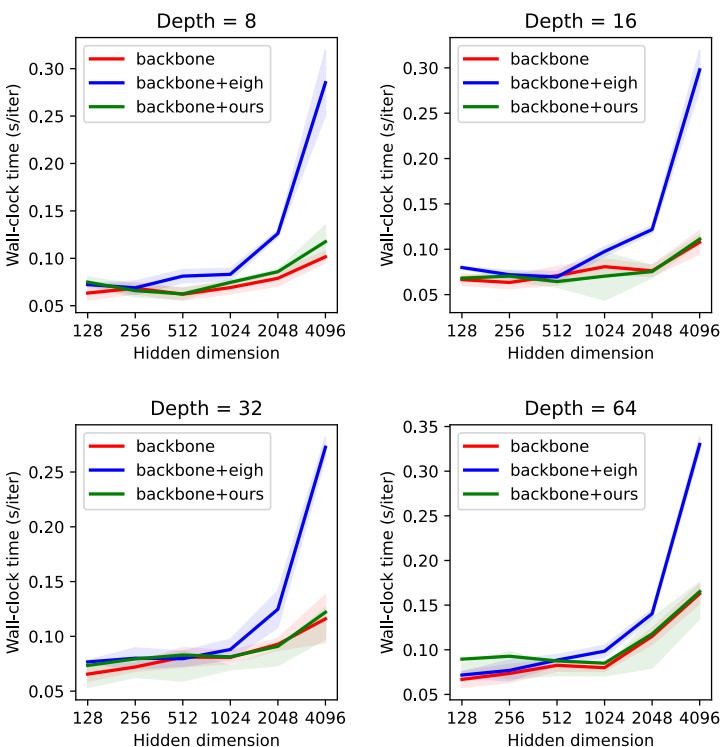

Figure 4: A comparison of scaling in the latent PCA task using the Celeb-A-HQ (256x256) dataset. The backbone autoencoders used in this study consist entirely of fully-connected layers with ReLU activation, all maintaining the same dimensions. Between the encoder and decoder, we applied both an eigen solver from the JAX `eigh` function and our amortized eigendecomposition method. We varied the depth of the autoencoders across 8, 16, 32, and 64 layers, and explored dimensionalities of 128, 256, 512, 1024, 2048, and 4096. The results present the average execution time per iteration over 100 runs. Notably, the largest model tested, featuring an autoencoder with 64 layers and a dimension of 4096, comprises up to 1.0 billion parameters.

additional term resembles Brockett's cost function while the trace of the covariance matrix ensures homogeneity:

$$\min_{\boldsymbol{\omega},\boldsymbol{\theta},\boldsymbol{W}} \left\| \mathsf{Dec}_{\boldsymbol{\omega}}\left(\boldsymbol{U}\boldsymbol{U}^T\mathsf{Enc}_{\boldsymbol{\theta}}(\boldsymbol{X})\right) - \boldsymbol{X} \right\|_{\mathrm{F}} - \eta \frac{\mathrm{tr}(\boldsymbol{M}\boldsymbol{U}^{\mathsf{T}}\mathrm{cov}(h_{\boldsymbol{\theta}}(\boldsymbol{X}))\boldsymbol{U})}{\mathrm{tr}(\mathrm{cov}(h_{\boldsymbol{\theta}}(\boldsymbol{X})))}, \qquad (19)$$

where cov represents the covariance function. The architecture for both the encoder and decoder mirrors that of the nuclear norm regularization model, with the exception that there is a linear projection aligning with the direction of the principal components. The average execution times are reported in Table 1. More results and analysis are provided in Appendix A.2. The experimental results demonstrate that our approach achieves an average training speed improvement of 40% compared to the conventional eigendecomposition approach.

Additionally, we conducted a scalability study, with the results presented in Figure 4. This study examined the scaling behavior of latent PCA on the Celeb-A-HQ (256x256) dataset [29] by varying both the depth and width of the backbone autoencoder, with average execution time per iteration reported. The largest model in our tests, a 64-layer autoencoder with a 4096-dimensional latent space, contains over 1 billion parameters. From these results, we draw two main conclusions. First, our amortized eigen loss substantially reduces the eigendecomposition training time without significantly increasing the computational load of the backbone, as evidenced by the close alignment of the red (backbone) and green (backbone + our approach) lines. In contrast, the traditional eigendecomposition approach (blue line) scales steeply with increasing dimensionality, whereas our approach exhibits a much slower growth rate. Second, eigendecomposition emerges as the primary computational bottleneck within these neural network architectures, while the fully-connected layer computation

remains minor, particularly at large widths (>2000). This is reflected in the widening gap between the `backbone` (red line) and `backbone + eigh` (blue line) as dimensionality increases. Notably, increasing the depth of the backbone while keeping the hidden dimension constant results in minimal change in execution time, indicating that the cost of fully-connected layers is small relative to eigendecomposition—further underscoring the results shown in Figure 1.

## 5.4 Adversarial Attacks on Graph Convolutional Networks

In this study, we explore the robustness of graph convolutional networks (GCNs) [21] by implementing adversarial attacks on graph structures. The objective of this problem is described in Eq. (5). Our approach simplifies the attainment of a low-rank structure by optimizing:

$$\min_{\boldsymbol{\theta}, \boldsymbol{W}} \quad \left\|\mathsf{GCN}_{\boldsymbol{\theta}}(\boldsymbol{X}, \widehat{\boldsymbol{L}}) - \boldsymbol{Y}\right\|_2 - \eta \mathsf{tr}(\boldsymbol{M}\boldsymbol{U}^\top \boldsymbol{L}\boldsymbol{U}), \tag{20}$$

where $\widehat{\boldsymbol{L}} = \boldsymbol{U}^\top \mathsf{diag}(\boldsymbol{U}^\top \boldsymbol{L}\boldsymbol{U})\boldsymbol{U}$, with $\boldsymbol{U}$ being parameterized by Eq. (13). The motivation for this formulation is that at optimum, the columns of $\boldsymbol{U}$ correspond to the top-k eigenvectors of $\boldsymbol{L}$. Then $\widehat{\boldsymbol{L}}$ becomes the best rank $k$ approximation of $\boldsymbol{L}$ under the Frobenius norm which corresponds to the terms $\left\|\widehat{\boldsymbol{L}}\right\|_*, \left\|\widehat{\boldsymbol{L}} - \boldsymbol{L}\right\|_F^2$ in Eq. (5). This formulation allows the GNN to operate on a low-rank graph $\widehat{\boldsymbol{L}}$, which has been shown to enhance robustness against adversarial attacks on the graph structure.

Our architecture consists of a three-layer GCN, which is utilized for semi-supervised node classification tasks on several citation networks, namely Cora, Citeseer, and Pubmed. Each layer has a hidden dimension of 32. The dropout rates are set to 0.4 for Cora and Citeseer, and to 0.1 for Pubmed, to prevent overfitting. For optimization, we employ the Adam algorithm with a learning rate of $10^{-3}$.

The adversarial attacks are executed by perturbing the graph structure through the random addition and deletion of edges in the adjacency matrix. We quantify the extent of these perturbations using a *contamination rate*, which is defined as the ratio of the altered edge count to the total node pairs.

Additionally, we propose a graph modification based on the original objective, as detailed in Eq. (5). The graph Laplacian is represented by a symmetric matrix, on which the eigh function is applied to compute the eigenvalues. The corresponding execution times are documented in Table 1. For a more comprehensive experiment results and analysis of our findings, please refer to the Appendix A.3.

## 6 Discussion and Conclusion

In this study, we address a class of deep learning problems that incorporate eigendecomposition within their constraints or objective functions. Such problems are prevalent in applications like nuclear-norm regularized denoising, network compression, graph structure learning, and whitening normalization. The traditional approach requires performing eigendecomposition or singular value decomposition within the computational graph, which becomes the bottleneck in the training process. To circumvent this computation overhead, we introduce an amortized eigendecomposition framework integrating a relaxation eigen loss into the learning objective, which relies on a set of orthonormal vectors. These vectors are reparameterized efficiently through QR decomposition, thereby substantially reducing the computational cost of eigendecomposition in each iteration. Furthermore, the differentiable nature of QR decomposition allows for its seamless incorporation into the neural network training workflow. Our experimental results demonstrate that, when applied to network tasks, our algorithm not only accelerates the process but also maintains the precision of the conventional method with eigendecomposition. The method proves particularly beneficial as a differentiable top $k$ eigensolver in environments where the backward gradient computation for top $k$ eigendecomposition, such as in JAX or PyTorch, is not well-supported. Consequently, our approach provides a viable alternative for integrating top $k$ eigendecomposition into neural networks.

**Limitations**. While our method excels in scenarios where eigendecomposition operations on large matrices are embedded within neural networks, it is important to note that when employed as a pure numerical eigensolver, it does not offer any speed or precision advantages over conventional methods. Thus, its suitability is specifically aligned with applications where eigendecomposition is a substantial component of the neural network training process.

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

# Appendix A    More Experimental Results

## A.1    Nuclear Norm Regularization

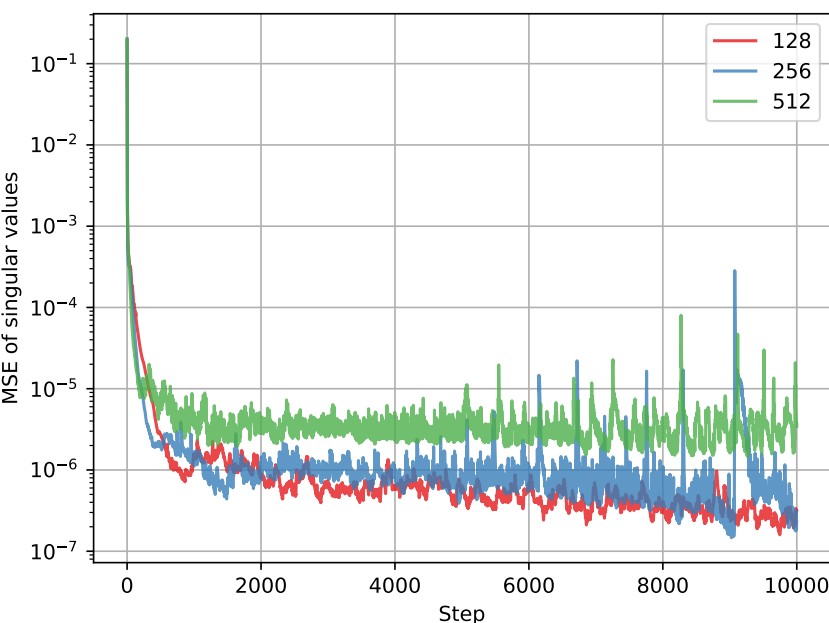

Figure 5: The convergence curve of the singular values.

In this experiment, we present the convergence analysis of our amortized eigendecomposition approach concerning the accurate estimation of singular values. We tackle the nuclear norm regularization problem outlined in Eq. (17). In each iteration, we compare the diagonal elements of $U^\top \theta^\top \theta U$ with the exact singular values of $\theta$. We conducted tests on $\theta$ of sizes $128 \times 128$, $256 \times 256$ and $512 \times 512$. The mean square error (MSE) is depicted in Figure 5.

The results indicate that initially, the singular values obtained by our approach do not match the correct singular values within a small error. However, within just a few iterations, our approach rapidly converges to the correct values, demonstrating the effectiveness of the loss function in accurately estimating singular values.

## A.2    Latent-space Principle Component Analysis

In this task, we evaluate the latent-space PCA method as outlined in Eq. (19) using the MNIST dataset. The architecture for both the encoder and decoder mirrors that of the nuclear norm regularization model, with the exception that there is a linear projection $UU^\top$ aligning with the direction of the principal components. We aim to extract the first two principal components in the latent space.

The experimental outcomes are depicted in Figure 6. Specifically, Figure 6 (a) displays the convergence trajectories for both the reconstruction loss and the eigenvalue loss as defined in Eq. (19). It is observed that the reconstruction loss curves for the conventional eigh function and our amortized eigendecomposition strategy are nearly indistinguishable. However, for the eigen-loss, our method initially registers lower values compared to the eigh function but eventually converges to equivalent values. This demonstrates the efficacy of the amortized optimization approach.

Figure 6(b) illustrates the distribution of features in the latent space along the two principal components for cases where $\eta = 0$ and $\eta = 1$. The eigenvalues correspond to the total variance across the feature dimensions. Without regularization (when $\eta = 0$), the scales of the principal component axes

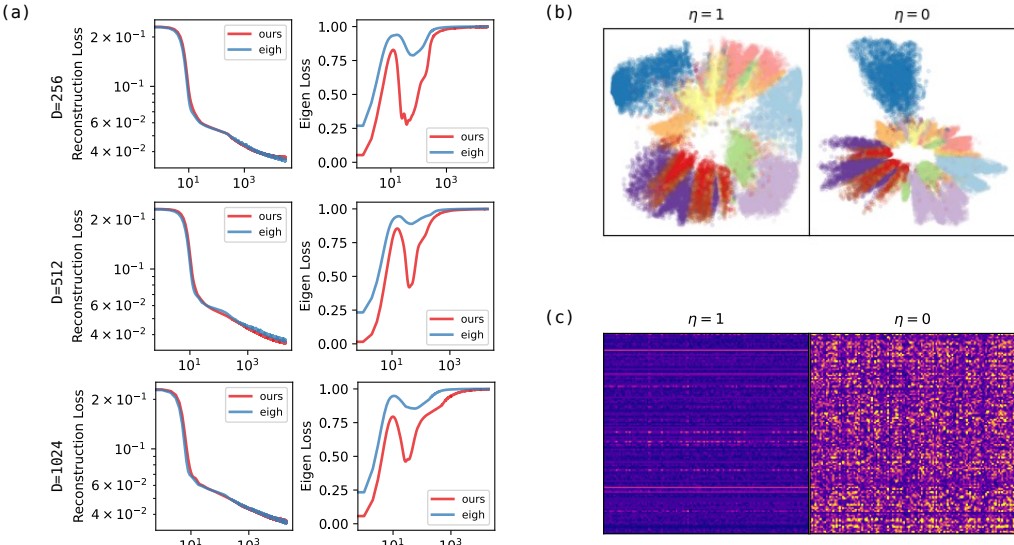

Figure 6: Experimental results of Latent-space PCA on MNIST dataset. (a) Convergence curves. First column: Convergence curve of reconstruction loss. Second column: Convergence curve of eigen loss. (b) Principle components in latent space: The two principle components of features in latent space for $\eta = 0$ and $\eta = 1$. (c) The sparsity of the network structure. The weight matrices of the second layer of the encoder. The color indicates the scale of the absolute values of the weight matrix ranging from 0 to 1.

can vary significantly. However, when regularization is applied (with $\eta = 0$), the distribution of the embedding becomes more uniform, indicating a more even spread across the principal components.

The regularization applied to the eigenvalues also influences the sparsity of the network's architecture. This effect is depicted in Figure 6(c), which shows the weight matrix of the last layer in the encoder. When $\eta = 0$, the network prioritizes learning weights that optimally reconstruct the images, resulting in a relatively dense weight configuration. Conversely, when $\eta = 0$, the eigenvalues corresponding to the principal components are encouraged to concentrate on the first few, leading to a decline in variance for the subsequent components. This, in turn, promotes sparsity in the weight matrix, as the network assigns less importance to the less variable components.

## A.3 Adversarial Attacks on Graph Convolutional Networks

This study investigates the robustness of graph convolutional networks (GCNs) [21] by conducting adversarial attacks on graph structures. These attacks involve perturbing the graph structure by randomly adding or removing edges in the adjacency matrix. To measure the magnitude of these perturbations, we introduce a *contamination rate*, which represents the ratio of the number of modified edges to the total number of node pairs.

For each adversarial scenario, we commence by randomly determining a *contamination rate* within the range of [0, 0.02] for the Cora and Citeseer datasets, and [0, 0.001] for the PubMed dataset. This rate dictates the proportion of edges to be randomly added or removed from the graph. A greater contamination rate signifies a more severe attack. The nodes are partitioned randomly into training, validation, and test sets with respective proportions of 60%, 20%, and 20%. The classification accuracy on the test set is then recorded.

The robustness of the GCN when subjected to graph contamination is illustrated in Figure 7. Each data point on the graph corresponds to a specific attack instance and its resultant test accuracy. Additionally, we have applied a polynomial regression with a 5th-degree basis to model the general trend of the relationship between the contamination rate and the test accuracy. The findings indicate that the graph convolutional network exhibits enhanced robustness when convolutions are applied to graph signals of a lower rank.

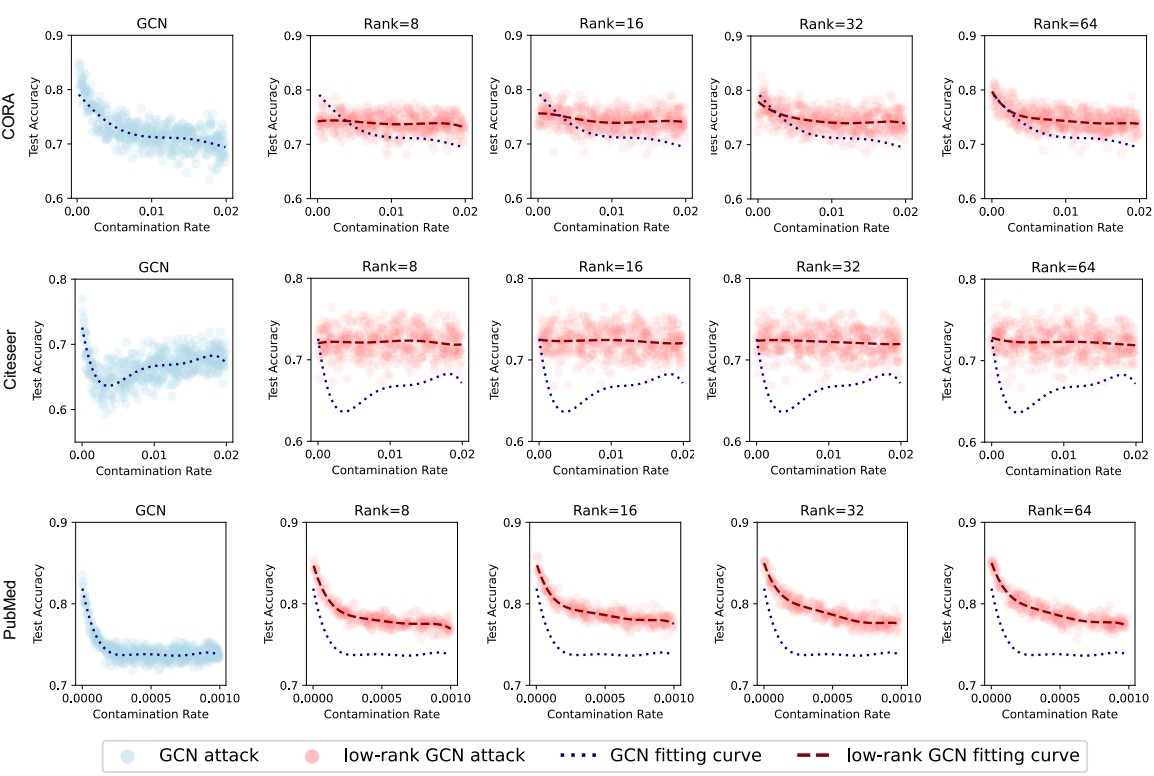

Figure 7: Experimental results of graph structure adversarial attacking

# Appendix B   Proofs

In this section, we provide the proofs for our theoretical findings.

## B.1   Preliminaries

**Lemma 1** (Jensen's inequality for convex functions). *For a convex function $f$, given $n$ real numbers $x_1, \ldots, x_n$ and non-negative weights $w_1, \ldots, w_n$ such that $\sum_{i=1}^{n} w_i = 1$, Jensen's inequality is expressed as:*

$$f\left(\sum_{i=1}^{n} w_i x_i\right) \geqslant \sum_{i=1}^{n} w_i f(x_i). \tag{21}$$

*The equality holds if and only if one of the following conditions is satisfied:*

1. *$f$ is linear;*

2. *$x_i$'s are equal;*

3. *one of $w_i$'s is one and the rest are zero.*

## B.2   Proof of Theorem 1

**Theorem 1** (Trace inequality with weight matrix) *Consider a $n$-dimensional symmetric matrix $\boldsymbol{A}$. Let $\boldsymbol{M}$ be a $k$-dimensional diagonal matrix with elements $0 < m_1 < m_2 \cdots < m_k$. For an arbitrary matrix $\boldsymbol{U} \in \mathbb{U}^{n \times k}$ with orthogonal columns, the following inequality always holds:*

$$\sum_{i=1}^{k} m_{k-i+1}\lambda_i \leqslant \text{tr}(\boldsymbol{M}\boldsymbol{U}^\mathsf{T}\boldsymbol{A}\boldsymbol{U}) \leqslant \sum_{i=1}^{k} m_i \lambda_i. \tag{22}$$

*The equalities are achieved if and only if $\boldsymbol{U}$ contains the eigenvectors corresponding to the $k$ largest (smallest) eigenvalues of $\boldsymbol{A}$.*

*Proof.* We can consider the case where $\boldsymbol{A}$ has all nonnegative eigenvalues $\varepsilon_j \geqslant 0$, without loss of generality, as we can always shift the diagonal elements of $\boldsymbol{A}$ to meet this condition:

$$\text{tr}\left(\boldsymbol{M}\boldsymbol{U}^\mathsf{T}\boldsymbol{A}\boldsymbol{U}\right) = \text{tr}\left(\boldsymbol{M}\boldsymbol{U}^\mathsf{T}\left(\boldsymbol{A} + \alpha\boldsymbol{I}\right)\boldsymbol{U}\right) - \alpha\sum_{i=1}^{k} m_i \tag{23}$$

Let $u_{ij}$ be the $i, j$-th entry of $\boldsymbol{U}$. The Right-hand-side (RHS) of the leftmost inequality can be expressed as,

$$\text{RHS} = \text{tr}(\boldsymbol{M}\boldsymbol{U}^\mathsf{T}\boldsymbol{A}\boldsymbol{U}) = \sum_{i=1}^{k}\sum_{l=1}^{n}\sum_{j=1}^{n} m_i a_{jl} u_{ji} u_{li} \tag{24}$$

We denote the actual eigenvectors of $\boldsymbol{A}$ as $\boldsymbol{V}$ with entries $\{\nu_{jl}\}$, and $\{\lambda_s\}, s = 1, \ldots, n$ as the eigenvalues, we have:

$$a_{jl} = \sum_{s=1}^{n} \lambda_s \nu_{js} \nu_{ls} \tag{25}$$

Substituting this into (24), we obtain:

$$\text{RHS} = \sum_{i=1}^{k}\sum_{l=1}^{n}\sum_{j=1}^{n} m_i \left(\sum_{s=1}^{n} \lambda_s \nu_{js} \nu_{ls}\right) u_{ji} u_{li} \tag{26}$$

$$= \sum_{i=1}^{k}\sum_{s=1}^{n} m_i \lambda_s \sum_{j=1}^{n} \nu_{js} u_{ji} \sum_{l=1}^{n} \nu_{ls} u_{li} \tag{27}$$

$$= \sum_{i=1}^{k}\sum_{s=1}^{n} m_i \lambda_s \left(\boldsymbol{\nu}_s^\mathsf{T}\boldsymbol{u}_i\right) \boldsymbol{\nu}_s^\mathsf{T}\boldsymbol{u}_i \tag{28}$$

We define

$$g_{si} = \boldsymbol{\nu}_s^\mathsf{T} \boldsymbol{u}_i \tag{29}$$

then the RHS becomes,

$$\text{RHS} = \sum_{i=1}^{k} \sum_{s=1}^{n} m_i \lambda_s g_{si}^2 \tag{30}$$

and,

$$0 \leqslant g_{si}^2 \leqslant 1. \tag{31}$$

It's worth noting that for any $s$,

$$\sum_{s=1}^{n} g_{si}^2 = \sum_{s=1}^{n} \boldsymbol{u}_i^\mathsf{T} \boldsymbol{\nu}_s \boldsymbol{\nu}_s^\mathsf{T} \boldsymbol{u}_i \tag{32}$$

$$= \text{tr} \left( \boldsymbol{u}_i^\mathsf{T} \boldsymbol{V} \boldsymbol{V}^\mathsf{T} \boldsymbol{u}_i \right) = 1 \tag{33}$$

Thus we have,

$$\text{RHS} = \sum_{i=1}^{k} m_i \left( \sum_{s=1}^{n} \lambda_s g_{si}^2 \right) \left( \sum_{s=1}^{n} g_{si}^2 \right) \tag{34}$$

Applying the Cauchy-Schwartz inequality, we get,

$$\text{RHS} \geqslant \sum_{i=1}^{k} m_i \left( \sum_{s=1}^{n} \sqrt{\lambda_s} g_{si}^2 \right)^2 \tag{35}$$

The equality holds if and only if for all $s$ and a fixed $l$,

$$\frac{\lambda_s g_{si}^2}{g_{si}^2} = \text{const.} \tag{36}$$

or

$$g_{si}^2 = 0 \tag{37}$$

This is only true when $g_{si}$ are either $0$ or $1$, with at most one non-zero value for each $s$ and $i$. In other words, $\boldsymbol{G} = \{g_{si}\}$ must be a permutation matrix. This implies that $\boldsymbol{U}$ is equivalent to the eigenvectors of $\boldsymbol{V}$, albeit with a possible permutation.

Thus this inequality reaches a minimum if we rearrange the indices of the eigenvalues:

$$\text{RHS} \geqslant \sum_{i=1}^{k} m_i \lambda_{\sigma(i)} \geqslant \sum_{i=1}^{k} m_{k-i+1} \lambda_i \tag{38}$$

The last inequality is a result of the rearrangement inequality. This completes the proof of the first inequality.

For the upper bound of Eq. (22), we apply the Cauchy-Schwartz inequality to Eq. (30) and obtain,

$$\text{LHS} \leqslant \sum_{i=1}^{k} m_i \sqrt{\sum_{s=1}^{n} \lambda_s^2 g_{si}^2 \sum_{s=1}^{n} g_{si}^2} \tag{39}$$

$$= \sum_{i=1}^{k} m_i \sqrt{\sum_{s=1}^{n} \lambda_s^2 g_{si}^2} \tag{40}$$

$$\leqslant \sum_{i=1}^{k} m_i \lambda_{\sigma(s)}. \tag{41}$$

Here again, the Cauchy-Schwartz inequality achieves the maximum when $U$ is equivalent to the eigenvectors of $V$, up to a possible permutation. By applying the rearrangement inequality, we further obtain the final bound:

$$\text{LHS} \leqslant \sum_{i=1}^{k} m_i \lambda_i. \tag{42}$$

This yields the desired upper bound for the second inequality of the main result. $\qquad \square$

### B.3 Proof of Theorem 2

**Theorem 2** (Trace inequality with convex function) *Let $A$ be a given n-dimensional symmetric matrix and let $U$ be a matrix of size $n \times k$ that resides on the Stiefel manifold. Suppose $f : \mathbb{R} \to \mathbb{R}$ be a monotonically increasing, convex function applied element-wise. The following inequalities hold:*

$$k \left( f \left( \frac{1}{k} \text{tr}(A) \right) \right) \leqslant \text{tr} \left( f(U^\top A U) \right) \leqslant \sum_{i=1}^{k} f(\lambda_i), \tag{43}$$

*The rightmost inequality becomes an equality if and only if $U$ comprises the eigenvectors of $A$ that correspond to the $k$ largest eigenvalues, The leftmost inequality is met with equality when $U$ is such that all diagonal elements of the matrix $U^\top A U$ are equal.*

*Proof.* **The rightmost inequality** We first focus on the rightmost inequality, which provides an upper bound for the convex trace loss.

To establish the rightmost inequality, let us consider the left-hand side (LHS) of the equation. Let's denote the diagonal elements of $A$ as $\{a_{jj}\}$. Consider the eigendecomposition of $A$ as in Eq. (25), we can rewrite $a_{jj}$ in terms of the eigenvalues and eigenvectors:

$$a_{jl} = \sum_{i=1}^{k} \lambda_i \nu_{ji} \nu_{li} \tag{44}$$

where $\nu_{ji}$ is the $(j, i)$ entry of its eigenvectors $V$. The diagonal elements of the matrix product $U^\top A U$ can then be written as:

$$\left( U^\top A U \right)_{ss} = \sum_{l=1}^{n} \sum_{j=1}^{n} a_{jl} u_{ls} u_{js} \tag{45}$$

Substituting this result back into the LHS, we have

$$RHS = \sum_{s=1}^{n} f \left( \left( U^\top A U \right)_{ss} \right) \tag{46}$$

$$= \sum_{s=1}^{n} f \left( \sum_{l=1}^{n} \sum_{j=1}^{n} a_{jl} u_{ls} u_{js} \right) \tag{47}$$

$$= \sum_{s=1}^{n} f \left( \sum_{l=1}^{n} \sum_{j=1}^{n} \sum_{i=1}^{k} \lambda_i \nu_{ji} \nu_{li} u_{ls} u_{js} \right) \tag{48}$$

$$= \sum_{s=1}^{n} f \left( \sum_{i=1}^{k} \lambda_i \sum_{l=1}^{n} \sum_{j=1}^{n} \nu_{ji} \nu_{li} u_{ls} u_{js} \right) \tag{49}$$

We can express the summation of the elements in vector form as follows:

$$\sum_{i=1}^{k} \sum_{l=1}^{n} \sum_{j=1}^{n} \nu_{ji} \nu_{li} u_{ls} u_{js} = \sum_{i=1}^{k} u_s^\top v_i v_i^\top u_s \tag{50}$$

$$= \text{tr}(u_s^\top V V^\top u_s) = 1 \tag{51}$$

For all indices $j$, $l$ and $s$, all the addends satisfy:

$$0 \leqslant \sum_{l=1}^{n} \sum_{j=1}^{n} \nu_{ji} \nu_{li} u_{ls} u_{js} \leqslant 1, \tag{52}$$

which allows us to treat $\sum_{l=1}^{n} \sum_{j=1}^{n} \nu_{ji} \nu_{li} u_{ls} u_{js}$ as a set of weights. By applying Jensen's inequality to the equation previously labeled as Eq. (49), we obtain:

$$LHS \leqslant \sum_{i=1}^{k} f\left(\lambda_i\right) \sum_{s=1}^{n} \sum_{l=1}^{n} \sum_{j=1}^{n} \nu_{ji} \nu_{li} u_{ls} u_{js} \tag{53}$$

Equality holds if and only if all the eigenvalues $\{\lambda_i\}$ are identical, which is generally not the case for an arbitrary symmetric matrix $\boldsymbol{A}$, or for each index $s$, there exists an index $i$ such that,

$$(\boldsymbol{u}_s^{\mathsf{T}} \boldsymbol{v}_i)^2 = 1, \tag{54}$$

with all other terms being zero. Therefore, $\boldsymbol{U}$ must be a permutation of $\boldsymbol{V}$:

$$\boldsymbol{u}_s = \boldsymbol{v}_{\sigma(i)} \tag{55}$$

Substituting this result back into Eq. (53), we obtain:

$$\sum_{i=1}^{k} f\left(\lambda_i\right) \sum_{s=1}^{n} \sum_{l=1}^{n} \sum_{j=1}^{n} \nu_{ji} \nu_{li} u_{ls} u_{js} \leqslant \sum_{i=1}^{k} f\left(\lambda_{\sigma(i)}\right) \leqslant \sum_{i=1}^{k} f\left(\lambda_i\right) \tag{56}$$

With this, we conclude the derivation of the first inequality.

**The leftmost inequality.** For the leftmost inequality,

$$RHS = \sum_{i=1}^{k} f\left(\sum_{l=1}^{n} \sum_{j=1}^{n} a_{jl} u_{li} u_{ji}\right) \tag{57}$$

$$= k \sum_{i=1}^{k} \frac{1}{k} f\left(\sum_{l=1}^{n} \sum_{j=1}^{n} a_{jl} u_{li} u_{ji}\right) \tag{58}$$

$$\geqslant k f\left(\frac{1}{k} \sum_{i=1}^{k} \sum_{l=1}^{n} \sum_{j=1}^{n} a_{jl} u_{li} u_{ji}\right) \qquad \text{(Jensen)} \tag{59}$$

$$= k f\left(\frac{1}{k} \operatorname{tr}\left(\boldsymbol{U}^{\mathsf{T}} \boldsymbol{A} \boldsymbol{U}\right)\right) \tag{60}$$

$$= k f\left(\frac{1}{k} \operatorname{tr}\left(\boldsymbol{A}\right)\right) \tag{61}$$

the equality holds if and only if $\sum_j u_{ij}^* u_{ij} a_{jj}$ equals for all $i$. $\qquad\square$

A similar result for finding the $k$ smallest eigenvalues can be obtained. The next corollary states the results

**Corollary** (Trace inequality with concave function) *Let $\boldsymbol{A}$ be a given n-dimensional symmetric matrix and let $\boldsymbol{U}$ be a matrix of size $n \times k$ that resides on the Stiefel manifold. Suppose $f : \mathbb{R} \to \mathbb{R}$ be a monotonically increasing, concave function applied element-wise. The following inequalities hold:*

$$\sum_{i=1}^{k} f(\lambda_i) \leqslant \operatorname{tr}\left(f(\boldsymbol{U}^{\mathsf{T}} \boldsymbol{A} \boldsymbol{U})\right) \leqslant k\left(f\left(\frac{1}{k} \operatorname{tr}(\boldsymbol{A})\right)\right), \tag{62}$$

*The rightmost inequality becomes an equality if and only if $\boldsymbol{U}$ comprises the eigenvectors of $\boldsymbol{A}$ that correspond to the $k$ smallest eigenvalues.*

# Appendix C  Further Extensions

## C.1  Related Work

**Eigensolver in numerical linear algebra**  Large-scale eigensolvers in numerical linear algebra usually involve two stages [39]. In the first stage, depending on whether the matrix is symmetric or not, either Arnoldi or Lanczos iteration is applied to reduce it into a matrix of smaller scale. Then, classical iteration methods such as power, inverse, and Rayleigh quotient iterations are applied to extract the eigensystem of the small-scale matrix. Compared to iterative solvers, our method is more easy to be integrated into the network training procedure.

**Eigendecomposition via manifold optimization**  In the optimization community, especially manifold optimization [1], identifying the eigen-subspace or solving for eigendecomposition is often achieved by optimizing the original trace loss function in Eq. 6 or the Brockett's cost function in Eq. 8 respectively. Instead of parametrizing the matrix $U$ via QR decomposition, one usually solves the constraint optimization via gradient descent with retraction. Namely, in each iteration, one firstly performs one step of gradient descent from a $U$ on the Stiefel manifold and then uses a retraction to correct it back to the manifold. Some convergence guarantees to the critical point and linear convergence rate can be established under certain assumptions [1, 43]. Various methods exist, such as [24] introducing the splitting method and [43] proposing a Crank-Nicolson-like update scheme to preserve the orthogonal constraints. No matter which method is used, the computation bottleneck lies in preserving the orthogonal constraints similar to our amortized method.

**Spectral normalization**  Spectral normalization is frequently added to improve the generalization ability of the neural network, i.e. CNN [11, 19] and GAN [32]. Compared to nuclear-norm regularization, spectral regularization only penalizes the largest absolute value of the eigenvalue so that the network outputs depend continuously on the input. Therefore no orthogonal constraint is enforced and the spectral norm can be more efficiently estimated based on the power iteration on a randomly initialized vector as in [32, 11].

## C.2  Extensions on the Amortized Eigendecomposition Approach

**Singular Value Decomposition Problem**  The singular value decomposition of a rectangular matrix $A \in \mathbb{R}^{m \times n}$ can be achieved by solving the following objective:

$$\max_{U \in \mathbb{U}^{n \times k}} \|AUM\|_{\mathrm{F}} \tag{63}$$

where $\| \cdot \|_{\mathrm{F}}$ represent the Frobenius norm. The $i$-th largest singular values can be obtained by $\sigma_i = \|Au_i\|_2$ with $u_i$ the $i$-th column of optimal $U$.

**Generalized Eigendecomposition Problem**  Generalized eigendecomposition is an extension of conventional eigendecomposition to a pair of matrices, rather than just a single matrix. The generalized eigendecomposition can be formulated as follows:

$$Au = \varepsilon Bu \tag{64}$$

where $A$ and $B$ are square matrices of same dimensions in the complex field $\mathbb{C}^{n \times n}$. The vector $u$ represents the eigenvector, and the scalar $\varepsilon$ denotes the corresponding eigenvalue. Together, $(\varepsilon, u)$ constitute the eigenpair of the generalized eigenvalue problem of the matrix pair $(A, B)$. In a manner akin to prior problems, the resolution to this generalized version can be achieved through the optimization of the following objective function:

$$\max_{U \in \mathbb{U}^{n \times k}} \sum_{i=1}^{k} m_i \frac{u_i^{\mathsf{T}} A u_i}{u_i^{\mathsf{T}} B u_i}. \tag{65}$$

where $m_i$'s are the weights as defined previously and $u_i$'s are the orthonormal column vectors of $U$.

