# OpenReview forum: "Amortized Eigendecomposition for Neural Networks"
_NeurIPS.cc/2024/Conference — NeurIPS 2024 poster_

### Official Review · Reviewer_fsNU · 2024-07-05

**Soundness:** 3
**Presentation:** 3
**Contribution:** 2
**Rating:** 5
**Confidence:** 3

**Summary:**

In this paper, the authors proposed a novel framework to integrate eigendecomposition efficiently into neural network training, which is typically costly in the backward process.

The key insight is to compute the decomposition together with training (since the decomposition also changes during training). Rather than forcing exact eigendecomposition $U\Sigma U^T=A_\theta$ every iteration, the authors treat the "eigenvectors" U as optimizable variables and leverage an eigen loss that attracts U to the true eigenvectors of $A_\theta$. The authors include a thorough analysis of different "eigen loss" inspired by the trace inequality (Eq.9, 11); they also provided an efficient implementation that re-parameterizes the orthogonal basis U into an unconstrained matrix $W$ via QR decomposition $U=\text{QR}(W)$.

For convergence analysis, the authors empirically showed that their eigen loss cooperates well with different optimizers for solving eigendecomposition; furthermore, they tested their method on several machine learning tasks and observed a significant speed-up from 1.4x to 600x.

**Strengths:**

Originality: Good. The idea of applying regularizers to simulate SVD/eigendecomposition has been proposed in previous works such as https://arxiv.org/abs/2303.10512. However, the framework proposed here is novel and detailed.

Quality: Good. The authors combined theoretical analysis (such as Fig.2) with empirical results (Fig.3).

Clarity: Good. The paper is well-written and easy to follow.

Significance: Fair. The experiments showed a substantial speedup for multiple tasks involving eigendecomposition.

**Weaknesses:**

The idea of doing eigendecomposition amortized is interesting. However, the stability and convergence rate of this method are not adequately discussed in the paper. Also, the design of eigen loss is very much "handcrafted", and I look forward to more flexible methods.

**Questions:**

1. It seems amortized eigendecomposition would converge slower w.r.t iterations, compared to traditional eigh/svd (because it's a rather indirect way of optimization). And I wonder how much time one can save for the whole training process?
2. Can this framework be applied to other aspects such as improving the stability of SVD in network training?

**Limitations:**

1. The proposed method seems to be designed for relatively simple machine learning tasks. But for large-scale deep learning tasks with complex loss landscapes, there would be many sub-optimal solutions and the eigen loss may not be helpful.

2. Limited to eigendecomposition which is only applicable to symmetric matrices. For example, in section 5.2 the authors have to approximate the nuclear norm $\|\theta\|_\star$ with $\sum_i \|\theta u_i\|$ since $\theta$ is not necessarily symmetric. So perhaps an "amortized SVD" would be more significant.

---

> ### Author Rebuttal · Authors · 2024-08-06
>
> We appreciate reviewer fsNU for the constructive feedback on our paper. Here are our responses.
>
> ***Q1 It seems amortized eigendecomposition would converge slower w.r.t iterations, compared to traditional eigh/svd (because it's a rather indirect way of optimization). And I wonder how much time one can save for the whole training process?***
>
> Our amortized eigendecomposition may indeed converge slower in terms of iterations compared to traditional eigh/SVD when used purely as a numerical eigensolver. We acknowledge this limitation in the conclusion section of the paper. However, the primary advantage of our approach lies in replacing the computationally expensive eigh/SVD with a more efficient QR decomposition. While the convergence to the same results may require more steps, the QR decomposition is significantly faster overall. In our experiments, we observed a substantial speed-up with our approach, ranging from 1.4x to 600x across various tasks.
>
> Additionally, we conducted an experiment on the latent PCA task using a backbone with 64 layers and over 1 billion parameters, and an eigendecomposition on a 4096x4096 matrix. The experimental results are shown in the additional PDF page. Our approach achieved at least a 50% reduction in time. Specifically, the average training time for such a backbone with eigh was 0.330 seconds per iteration, whereas our method reduced this to 0.155 seconds per iteration.
>
> ***Q2 Can this framework be applied to other aspects such as improving the stability of SVD in network training?***
>
> Yes, our method can enhance the stability of SVD during network training. QR decomposition tends to be more stable than SVD, especially when the underlying matrix $A$ is full-ranked [see 1]. This stability can be achieved through effective initialization of $W$ (where $U = QR(W)$ ), and the rank of $W$ is preserved during optimization on the orthogonal manifold.
>
> Furthermore, our approach can serve as a robust alternative to truncated SVD (ie. we just want the top k singular values), which is often not differentiable in frameworks like JAX and TensorFlow, or lacks numerical stability, as seen with methods like `torch.lobpcg`. Our method provides a stable and differentiable approach to matrix decompositions, making it a viable option in these scenarios.
>
> ***Q3 For large-scale deep learning tasks with complex loss landscapes, there would be many sub-optimal solutions and the eigen loss may not be helpful.***
>
> We want to clarify that the primary contribution of this paper is not to claim that the eigen loss itself improves neural network performance. Instead, our focus is on accelerating neural networks that involve eigendecomposition (or SVD) operations. Our approach aims to make these operations faster without compromising the accuracy of the eigendecomposition results. While it is true that involving eigendecomposition can make the loss landscapes more complex, there is extensive research demonstrating its effectiveness in various tasks, such as nuclear norm regularization, graph convolutional networks, and network compression. Our paper’s goal is to provide a more efficient way to perform eigendecomposition within neural networks, achieving speed improvements in an amortized fashion while preserving performance.
>
> ***Q4 Limited to eigendecomposition which is only applicable to symmetric matrices.***
>
> Although this paper focuses on eigendecomposition, our approach is NOT limited to symmetric matrices. The singular values of an asymmetric matrix $A$ can be obtained by performing eigendecomposition on the matrix $A^\top A$ (or $AA^\top$) and taking the square root of the eigenvalues, which is a well-known result. In Section C2 of the appendix, we demonstrate the extension of our approach to SVD. However, we chose to focus on eigendecomposition in this  paper because it is a more fundamental and more commonly-used linear algebra operation compared to SVD. For the next version of this manuscript, we will make this point more clear.
>
> [1]. Demmel, James W. Applied numerical linear algebra. Society for Industrial and Applied Mathematics, 1997.

---

> > ### Comment · Reviewer_fsNU · 2024-08-08
> >
> > Thanks for the comprehensive rebuttal, especially **Q3** illustrating that the eigen loss can act as a regularizer and preserve networks' performance. I am increasing my rating to 5.

---

### Official Review · Reviewer_WiHy · 2024-07-08

**Soundness:** 4
**Presentation:** 4
**Contribution:** 4
**Rating:** 8
**Confidence:** 3

**Summary:**

This paper proposes a method that decouples the eigendecomposation calculation from training process, instead, they use a eigen loss to jointly optimize it with the training loss of the neural network as a nested optimization loop. The proposed method can speed up the training process of problems that incorporate eigendecomposition within their constraints or objective functions.

**Strengths:**

The overall method is easy to understand and soundness. Combining the designed eigen loss with normal training loss is an interesting and reasonable method to effectively speed up the training process.

**Weaknesses:**

The theoretical support of this paper is sufficient. However, in the experiment section, except for the speed up ratio, I think the author should report the loss value and the corresponding task performance for a comprehensive comparison. It is important to verify whether the accelerate method can maintain the performance.

**Questions:**

I have no questions.

**Limitations:**

The author clearly state the limitation of this paper.

---

> ### Author Rebuttal · Authors · 2024-08-06
>
> We would like to thank reviewer WiHy for the constructive comments on our paper. Here are our responses.
>
> ***Q1 Whether the accelerate method can maintain the performance.***
>
> In the manuscript, we try to answer this question through two experiments:
> - Latent-space Principal Component Analysis: As shown in Figure 5a, the convergence curves of the traditional eigendecomposition and our approach align well. The reconstruction loss curves for both the conventional eigh function and our amortized eigendecomposition strategy are nearly indistinguishable. Initially, our method registers lower eigen-loss values compared to the eigh function, but it eventually converges to equivalent values. This demonstrates the efficacy of the amortized optimization approach.
> - Adversarial Attacks on Graph Convolutional Networks: As shown in Figure 6, we find that incorporating eigendecomposition of the Laplacian matrix makes the graph convolutional network more robust to attacks on graph structures.
>
> Both experiments demonstrate that our acceleration method can maintain performance without any degradation. Due to the page limit of the review round, we were not able to include these results in the main context. We make this point more clear in the main context of the paper if additional page is allowed.

---

> > ### Comment · Reviewer_WiHy · 2024-08-12
> >
> > Thanks for your detailed explanation, I will increase my score to 8.

---

### Official Review · Reviewer_5kBi · 2024-07-08

**Soundness:** 4
**Presentation:** 4
**Contribution:** 4
**Rating:** 8
**Confidence:** 4

**Summary:**

This paper proposes a method named "amortized eigendecomposition" to replace the computationally costly SVD operation in settings where an eigendecomposition is required during neural network training. The proposed method introduces a loss term ("eigen loss") and replaces the full SVD with the less computationally expensive QR decomposition at each iteration. A theoretical analysis shows that the desired eigenpairs are obtained as optima of the eigen loss. An empirical analysis on several tasks, including nuclear norm regularization, latent-space PCA, and graph adversarial learning, demonstrates that the proposed method attains significant improvements in training efficiency while producing nearly identical outcomes to conventional approaches.

**Strengths:**

Excellent organization and presentation, reads like a textbook chapter.
The method is well motivated and clearly explained. The analyses are on-point and thorough.
The significance and applications of the method is made clear, and the limitations are adequately discussed.
The method addresses a concrete application and is of immediate benefit for a class of problems in ML.

**Weaknesses:**

A discussion of scalability and comparison with alternative methods for fast SVD are two significant points that are missing in this paper. The experiments report results on small-to-moderate matrix dimensions.

**Questions:**

How does amortized eigendecomposition scale to larger dimensions?
How does it compare to other fast-SVD methods? In particular, I have randomized SVD in mind, which is known to scale very well. Is there a trade-off, or is there a reason for why randomized SVD is not applicable in this setting? Or is it just still more computationally expensive because it needs to be done in each iteration?

**Limitations:**

The authors have addressed the limitations of their method reasonable well. As mentioned in the sections above (weaknesses/questions), an additional discussion of scalability and comparison to other fast/approximate SVD methods would be nice to have.

---

> ### Author Rebuttal · Authors · 2024-08-06
>
> We would like to thank reviewer 5kBi for his acknowledgement and valuable comments of our paper. Here are our responses.
>
> ***Q1 Regarding the scalability.***
>
> For additional experiments on large-scale setups, please refer to the general response provided to all reviewers.
>
> ***Q2 Comparison with randomized SVD.***
>
> While randomized SVD can enhance the efficiency of SVD by randomly projecting the matrix into a smaller one and then performing SVD on the smaller projected matrix, there are several key differences between our approach and randomized SVD:
> - *Applicability*: Randomized SVD is more suitable for large **sparse** matrices with **low rank**, as mentioned in the document of `torch.svd_lowrank`. Our method, on the other hand, is a general eigendecomposition/SVD technique that makes no assumptions about the input matrix. For dense matrices, randomized SVD often suffers from lower performance compared to standard SVD. As shown in Figures 3 and 4, our approach achieves error rates close to those of standard SVD on dense matrices.
> - *Determinism*: Randomized SVD is a stochastic algorithm, meaning the results can vary due to inherent randomness. Although this variability can be mitigated by using a fixed random seed, it still introduces a non-deterministic element to the computation. In contrast, our algorithm is deterministic, providing consistent and reproducible results.
> - *Hyperparameter*: The accuracy of the randomized SVD approximation depends on the choice of the oversampling parameter p. If p is too small, the approximation may be poor. Conversely, if p is too large, it reduces computational efficiency. Our method does not require any specification of hyperparameters.
>
> We acknowledge that our algorithm could be further accelerated using randomized linear algebra techniques such as randomized QR decomposition or randomized trace estimators. We plan to explore these possibilities in future work.

---

### Official Review · Reviewer_6W8h · 2024-07-12

**Soundness:** 4
**Presentation:** 4
**Contribution:** 3
**Rating:** 7
**Confidence:** 4

**Summary:**

The paper describes a novel method to circumvent the need for explicit eigendecomposition during neural network training. The central insight is that one can simply learn estimates of the eigenvectors (parameterized by a QR decomposition) of interest alongside the original loss function via gradient descent.
To do so the author's add an auxiliary (and differentiable) loss whose optima is obtained by the appropriate set of eigenvectors.
A summary of the presented results:
- The author's provide both theoretical (proofs) and empirical evidence that the proposed estimation algorithm converges to the true eigenvectors.
- 3 settings where SVD or eigendecomposition are incorporated in neural network training schemes are investigated empirically, in all settings the proposed scheme yields a significant speed up relative explicit matrix decompositions.

**Strengths:**

- Motivation: The paper benefits from a very clear goal: to reduce the computational load of neural network training schemes that require eigendecomposition in each iteration. While this may be a somewhat niche use case, I believe such approaches are becoming increasingly prevalent. For example besides the three situations considered in the paper multiple recent self-supervised learning methods for learning image representations could benefit from the proposed speedups [1, 2].

- Novelty: The idea of learning estimates of eigenvectors using loss functions from existing optimization methods for eigendecomposition in neural network training is novel to the best of my knowledge. The author's discuss existing solutions in the appendix C. (iterative solvers, manifold optimization, etc.) and it seems clear to me the proposed method fills a hole in the practitioner's toolbox. I think this section could potentially be promoted to the main text if space permitted.

- Clarity of presentation: The theorems and their proofs, the description of the algorithm, and the experimental design are all clearly presented and easy to understand.

- Strong empirical results: In all of the considered settings the proposed amortization method provides significant speedups relative to full eigendecomposition (or SVD) during each iteration. I would be curious to know how such benefits scale as larger scale networks are employed (see below). Additional results in the appendix also suggest the amortization method results in similar solutions to the traditional approach.


[1] Ermolov, Aleksandr, et al. "Whitening for self-supervised representation learning." International conference on machine learning. PMLR, 2021.

[2] Yerxa, Thomas, et al. "Learning efficient coding of natural images with maximum manifold capacity representations." Advances in Neural Information Processing Systems 36 (2023): 24103-24128.

**Weaknesses:**

- Small scale experiments: Each of the three empirical experiments are performed with relatively small networks and datasets. While this is not necessarily a weakness in and of itself, my intuition is that the speed-up benefits will be more marginal as network size increases. This is because, for example as the depth/width of the encoder/decoder increases in the latent space PCA setting, the eigendecomposition becomes less of a bottleneck as more time is spent propogating activations through the network. If this is the case I think it merits discussion in the paper. On the same note it would be nice to have more assurances that the amortization method finds correct solutions (a la Figure 5a) on larger scale problems.

**Questions:**

- I would appreciate if the author's could comment on the scalability of the method (or more precisely, how the computational benefits scale with the architecture size).

- I found myself slightly confused by Eq. 19. The numerator of the regularizer encourages high variance along the first two PCs (and synergistically encourages $U$ to converge toward the first two PCs). Doesn't this, alongside the projection and reconstruction loss, encourage the covariance matrix to be low rank (as any variance in lower variance eigenspaces will not be accessible to the deocder)? Does homogeneity need to be preserved simply because without trace normalization the weights of the encoder would grow large and the weights of the decoder would decay? Sorry for getting caught up in this as it really is a detail, but wouldn't this also be a case where we really only care that $U$ forms a basis for the space of the top 2 PCs (because the first linear layer of the decoder could "undo" any such rotation of the space if it were advantageous)? If this is the case, I assume then it would be sufficient to set $M=I$?

**Limitations:**

Yes.

---

> ### Author Rebuttal · Authors · 2024-08-06
>
> We thank reviewer 6W8h for the constructive comments and valuable suggestions. Here are our responses.
>
> ***Q1 Regarding the scalability.***
>
> For additional experiments on large-scale setups, please refer to the general response provided to all reviewers.
>
> We would like to emphasize that, as shown in Figure 1, eigendecomposition/SVD is significantly more expensive than matrix multiplication—potentially up to 1000 times slower. If a network increases in depth while keeping the width constant, the additional matrix multiplications do not significantly increase computational cost. However, if the width of the network increases substantially, the cost of eigendecomposition operations escalates dramatically. For example, in the experiment discussed in the general response, the training time of an MLP model with 64 layers and over 1 billion parameters is still faster than performing an eigendecomposition operation on a 4096x4096 matrix. Thus, the eigendecomposition typically becomes the bottleneck unless the backbone is a significantly large neural network.
>
> ***Q2 Regarding Eq. 19.***
>
> Thank you for highlighting this confusion. Yes, the loss function in Eq. 19 encourages the covariance matrix to be low rank, with the rank being close to or equal to the number of principal components (PCs). Additionally, If the trace normalization is removed, the eigenvalues of the eigen loss can cause the output of the encoder to become excessively large.
>
> If we set $\mathbf{M}=\mathbf{I}$, $\mathbf{U}$ can find the optimal subspace, but the resulting PCs are not orthogonal, meaning the covariance matrix of the PCs is not diagonal. Therefore, $\mathbf{M}$ must have distinct diagonal elements.
>
> We acknowledge that this loss function may not be the clearest way to demonstrate latent PCA. The purpose of introducing this trace-normalized eigen loss (Eq. 19) is to demonstrate that a simple PCA layer in a network can enforce sparsity in the neural network, as shown in Figure 5c. However, we also find that this loss function may cause some confusion. An alternative loss is to avoid concerns about the eigenvalues by using a stop-gradient operator, as in Eq. 15, applied to the covariance matrix of the hidden output. The eigenloss would then become:
>
> $$ tr(\mathbf{M}\mathbf{U} \text{StopGrad} (\text{cov} (h_\theta(X)) \mathbf{U} ) $$
>
> In this case, the optimal $\mathbf{U}$ would be the PCA projection, and the trace loss would no longer cause the encoder/decoder parameters to become excessively large or small. Just, sparsity would not be enforced directly. We will modify this loss in the manuscript.

---

> > ### Comment · Reviewer_6W8h · 2024-08-12
> > **Response to Rebuttal.**
> >
> > Thank you to the author's for their clarifying comments. I keep my original score.

---

### Author Rebuttal · Authors · 2024-08-06

We sincerely thank all reviewers for their in-depth comments and valuable suggestions, which have significantly improved the quality of our paper. As many of the reviewers mentioned the scalability of our approach, we would like to provide a general response on this aspect.

We conducted an additional study on the scalability, with the results presented in the additional PDF page. Using the Celeb-A-HQ (256x256) dataset, we examined the scaling of the latent PCA task by varying the depth and width of the backbone autoencoder. The average execution time per iteration is reported. Notably, the largest model tested, with an autoencoder of 64 layers and a dimension of 4096, comprises over 1.0 billion parameters.

From the results, we can draw two main conclusions:

- **Efficiency of amortized eigen loss**: Our amortized eigen loss does not significantly increase the computational cost of the backbone but greatly reduces the training time for eigendecomposition. This is evident from the close alignment of the red (backbone) and green (backbone + our approach) lines. Compared to the traditional eigendecomposition approach (shown in the blue line), which increases dramatically as the dimension increases, our approach scales much more slowly.

- **Bottleneck of the latent PCA**: The primary bottleneck in such neural network structures is the eigendecomposition, while the computation for fully-connected layers is comparatively minor, especially when the width is large (>2000). This is demonstrated by the increasing gap between the backbone (red line) and backbone + eigh (blue line) as the dimension increases. However, if we increase the depth of the backbone while keeping the hidden dimension fixed, the execution time remains relatively unchanged. This indicates that the cost of fully-connected layers is small compared to that of eigendecomposition, echoing the results shown in Figure 1.

It is important to note that the total execution time includes both the computation of the eigen solver and the backbone autoencoders. The speed-up ratios presented in Table 1 reflect the speed-up of the eigen solvers alone, excluding the computation time of the backbone autoencoders.

---

> ### Comment · Reviewer_5kBi · 2024-08-12
> **ok**
>
> thanks for adding this info

---

### Decision · Program_Chairs · 2024-09-25

**Decision:**

Accept (poster)

**Comment:**

While the use case is somewhat niche (for neural networks relying on eigendecomposition during training), the reviewers found the paper clear, the idea novel and well supported by the experimental setting. The reviewers raised concerns regarding the scalability of the method that were addressed by the authors in their rebuttal. I recommend for acceptance.